# Performance of Nuclear Magnetic Resonance-Based Estimated Glomerular Filtration Rate in a Real-World Setting

**DOI:** 10.3390/bioengineering10060717

**Published:** 2023-06-13

**Authors:** Amauri Schwäble Santamaria, Marcello Grassi, Jeffrey W. Meeusen, John C. Lieske, Renee Scott, Andrew Robertson, Eric Schiffer

**Affiliations:** 1Department of Research and Development, Numares AG, 93053 Regensburg, Germany; 2Department of Laboratory Medicine and Pathology, Mayo Clinic, Rochester, MN 55905, USA; 3Division of Nephrology and Hypertension, Mayo Clinic, Rochester, MN 55905, USA

**Keywords:** glomerular filtration rate, eGFR, mGFR, GFR_NMR_ equation, CKD-EPI_2021Cr_ equation, CKD-EPI_2021CrCys_ equation, NMR, chronic kidney disease, CKD, routine sample validation

## Abstract

An accurate estimate of glomerular filtration rate (eGFR) is essential for proper clinical management, especially in patients with kidney dysfunction. This prospective observational study evaluated the real-world performance of the nuclear magnetic resonance (NMR)-based GFR_NMR_ equation, which combines creatinine, cystatin C, valine, and myo-inositol with age and sex. We compared GFR_NMR_ performance to that of the 2021 CKD-EPI creatinine and creatinine-cystatin C equations (CKD-EPI_2021Cr_ and CKD-EPI_2021CrCys_), using 115 fresh routine samples of patients scheduled for urinary iothalamate clearance measurement (mGFR). Median bias to mGFR of the three eGFR equations was comparably low, ranging from 0.4 to 2.0 mL/min/1.73 m^2^. GFR_NMR_ outperformed the 2021 CKD-EPI equations in terms of precision (interquartile range to mGFR of 10.5 vs. 17.9 mL/min/1.73 m^2^ for GFR_NMR_ vs. CKD-EPI_2021CrCys_; *p* = 0.01) and accuracy (P15, P20, and P30 of 66.1% vs. 48.7% [*p* = 0.007], 80.0% vs. 60.0% [*p* < 0.001] and 95.7% vs. 86.1% [*p* = 0.006], respectively, for GFR_NMR_ vs. CKD-EPI_2021CrCys_). Clinical parameters such as etiology, comorbidities, or medications did not significantly alter the performance of the three eGFR equations. Altogether, this study confirmed the utility of GFR_NMR_ for accurate GFR estimation, and its potential value in routine clinical practice for improved medical care.

## 1. Introduction

Glomerular filtration rate (GFR) is a critical clinical parameter in routine medical care and for the diagnosis and monitoring of kidney diseases [1]. To minimize systematic errors among and between patient groups, equations for estimated GFR (eGFR) incorporate demographic variables such as sex, age, and—until recently—race, together with endogenous filtration markers such as creatinine and cystatin C. As race is a social construct [2], healthcare professionals called for a reassessment of the inclusion of race in eGFR equations. In response, the Chronic Kidney Disease Epidemiology Collaboration (CKD-EPI) recently published the 2021 CKD-EPI eGFR equations that do not incorporate race [3]. Several studies demonstrated the clinical impact of implementing the new race-free equations on global patient management [4], and its benefit for kidney transplantation eligibility listing for Black patients [5,6,7,8].

The National Kidney Foundation (NKF) and the American Society of Nephrology (ASN) Task Force recently recommended the implementation of the race-free 2021 CKD-EPI creatinine equation (CKD-EPI_2021Cr_) for US adults [9,10]. They also recommended facilitating the routine use of cystatin C, because equations combining creatinine and cystatin C (e.g., CKD-EPI_2021CrCys_) are more accurate [9,10]. Finally, the Task Force encouraged the development of accurate, unbiased, precise, race-free equations integrating new endogenous filtration markers, to promote health equity [9,10]. We recently described GFR_NMR_, a nuclear magnetic resonance (NMR)-based equation without a race variable, combining the serum biomarkers creatinine, cystatin C, valine, and myo-inositol [11,12]. GFR_NMR_ demonstrated lower bias, and higher accuracy and precision than the 2009 and 2012 CDK-EPI equations [11], and demonstrated analytical performance suitable for routine clinical use [12]. When compared with the 2021 creatinine CKD-EPI_2021Cr_ equation, GFR_NMR_ showed comparable bias and significantly higher P15 accuracy in patients with or without a kidney transplant, and a stronger agreement with CKD staging by measured GFR (mGFR) in kidney transplant recipients [13]. Therefore, GFR_NMR_ may fulfill the recommendation of the NKF-ASN Task Force and hold promise for an alternative, well-performing and equitable eGFR equation.

Given the importance of accurately estimating GFR in medical care, particularly for proper CKD staging and clinical decision making, there is a clear need for independent validation of GFR_NMR_ in a real-world setting to confirm its potential use for improved patient management. Therefore, the aim of this prospective observational study was to validate GFR_NMR_ in a setting of routine clinical practice. Blood was collected from patients scheduled for clinically indicated urinary iothalamate clearance measurement, and GFR was estimated in fresh sera using GFR_NMR_ and the guideline-recommended CKD-EPI_2021Cr_ and CKD-EPI_2021CrCys_ race-free equations. A wide range of clinical data describing the etiology, comorbidities, and medications of the study sample were collected to investigate their potential impact on eGFR results.

## 2. Materials and Methods

### 2.1. Study Design and Participants

A prospective, observational, single center (Mayo Clinic, Rochester, MN, USA) study was conducted in 120 patients ≥ 18 years old scheduled for urinary iothalamate clearance measurement between May and September 2022. During the recruiting period, all patients scheduled for mGFR as part of clinical routine were screened for eligibility. Patients under hemodialysis or under peritoneal dialysis within seven days before urinary iothalamate clearance measurement were excluded from enrolment. Some patients were recruited repeatedly in the study, and the results considered as independent. Patients’ demographic and clinical data (including comorbidities and medications) were documented on the day of examination. 

The aim of the study was to validate GFR_NMR_ in the actual daily routine setting of clinical practice. The clinical reference standard was GFR measured by urinary iothalamate clearance, which is part of standard care at Mayo Clinic (Rochester, MN, USA). Fresh blood samples were collected immediately prior to mGFR measurement to estimate GFR (eGFR) using the NMR-based GFR_NMR_ equation [11,12] and the guideline-recommended 2021 CKD-EPI equations (CKD-EPI_2021Cr_ and CKD-EPI_2021CrCys_) [3,9,10]. Physicians and patients were blinded to the eGFR results.

The study was conducted according to the guidelines of the Declaration of Helsinki and approved by the respective Institutional Review Board (Mayo Clinic IRB #21-007723, dated 11 October 2021). All patients gave written informed consent prior to enrolment.

### 2.2. Sample Collection and Storage

A total of 120 blood samples (9 mL) were collected by venipuncture immediately before injection of iothalamate to measure GFR by urinary iothalamate clearance. Whole blood was allowed to clot for 30–120 min at room temperature and was centrifuged to collect blood serum. Fresh sera (at least 3 mL) were aliquoted (two 1 mL and two 0.5 mL aliquots) and stored at 4 °C until testing. On the day of examination, one 0.5 mL aliquot was used to measure creatinine and cystatin C levels, as described in Section 2.3. Then, one 1 mL aliquot was used within four days of blood collection for NMR measurement, as described in Section 2.3.

### 2.3. Laboratory Methods

GFR was measured at the Mayo Clinic Renal Testing Laboratory (Rochester, MN, USA) by urinary iothalamate clearance (non-radiolabeled) using liquid chromatography-tandem mass spectrometry, as previously reported [13,14]. The mGFR was normalized to body surface area according to the Dubois equation (body surface area = height^0.725^ × weight^0.425^ × 0.007184) and expressed as milliliter per minute per 1.73 m^2^ body surface area (mL/min/1.73 m^2^).

Biomarker measurements were performed on fresh refrigerated sera at Central Clinical Chemistry Laboratory, Mayo Clinic, Rochester, MN, USA. Serum creatinine was measured by enzymatic assay standardized to international reference materials [15], using Roche Cobas clinical analyzers (c701 or c501, Roche Diagnostics, Indianapolis, IN, USA). Cystatin C was measured by an immunoturbidometric assay (Gentian ASA, Moss, Norway) that was traceable to an international reference material [16], using a Roche Cobas c501 analyzer (Roche Diagnostics; Indianapolis, IN, USA).

Serum creatinine, valine, and myo-inositol were measured by NMR spectroscopy as previously described [11,13,17]. Briefly, 540 μL serum was mixed with 60 μL of Axinon^®^ serum additive solution and 600 μL was transferred into a 5 mm NMR tube with a barcoded cap. Samples were pre-heated at 37 °C for 7.5 min before NMR measurement in a Bruker Avance III 600 MHz, and a 5 mm PATXI probe equipped with automatic Z gradients shimming. The ^1^H-NMR spectra were recorded using a spectral width of 20 ppm, with a recycling delay of 1.5 s, 16 scans, and a fixed receiver gain of 50.4. A cycling time d2 of 8 ms was used together with a corresponding T2 filter of 112 ms. The mixing time τ between two consecutive spin echoes was 400 μs. The NMR data were automatically phase- and baseline-corrected using the lactate doublet at 1.32 ppm as reference. Metabolite quantification used curve-fitted pseudo-Voigt profiles, as previously described [11,17]. In case of analysis failure, the second 1 mL aliquot was used to repeat the NMR analysis. In case of repeated failure, the sample was excluded from final analysis.

GFR was estimated (eGFR) via three methods: GFR_NMR_, CKD-EPI_2021Cr_, and CKD-EPI_2021CrCys_. GFR_NMR_ test results were automatically generated by the Axinon^®^ NMR software (Numares AG, Regensburg, Germany), integrating age, sex, cystatin C (immunoturbidometric assay) and NMR measurements of creatinine, myo-inositol, and valine. The 2021 CKD-EPI equations were calculated within R using the reported formulas [3], combining either age, sex, and creatinine (enzymatic assay) for CKD-EPI_2021Cr_, or age, sex, creatinine (enzymatic assay), and cystatin C (immunoturbidometric assay) for CKD-EPI_2021CrCys_.

### 2.4. Statistical Analysis

Sample size was estimated based on existing GFR_NMR_ P20 accuracy results using the MedCalc Statistical Software version 12.7.7, according to Machin et al. [18]. Based on an estimated error not exceeding 15% absolute, with a two-sided alpha level of 5% and a power of 90%, a minimum of 106 patients was required for the study. Including a safety margin of 15% for possible dropouts, a total of 120 patients was planned for enrolment.

Performance evaluation (signed median bias, precision, accuracy, and precision intervals) was conducted in all enrolled patients with a valid GFR_NMR_ result. Subgroup analyses were also conducted to evaluate the impact of disease, comorbidities, or medication on the performance of each eGFR equation.

All statistical evaluations were performed within R 4.0.2 [19]. Data structures were handled with data.frame [20], data.table [21], and archivist [22] packages. Bootstrap procedures were implemented via the boot package [23,24]. Visualization was performed with ggplot2 [25]. In bias, precision and accuracy analyses, the respective 95% confidence intervals (CI) were calculated using the bootstrap method. In all analyses, *p*-values ≤ 0.05 were considered statistically significant.

Bias was calculated as ‘eGFR-mGFR’ and expressed as median signed bias to mGFR. Pairwise significance levels between bias distributions were assessed via the Wilcoxon-signed rank test [26,27] with *p*-value correction for multiple testing according to Benjamini-Hochberg [28,29].

Precision was assessed by the interquartile range (IQR) of the difference to mGFR. Significance of differences was assessed via the bootstrap method. 

Accuracy was evaluated by the percentage of samples with an eGFR within 15% (P15), 20% (P20), or 30% (P30) of mGFR. Pairwise comparisons were tested using the McNemar’s Chi-squared test [30] and Benjamini-Hochberg correction [28,29].

Distribution of mGFR at any given eGFR was assessed by fitting a quantile regression model for quantiles 2.5th, 10th, 25th, 50th, 75th, 90th, and 97.5th (one model for each quantile value, for a total of 7 quantile regression models). Quantile regression differs from the ordinary least squares (OLS) regression in that OLS regression estimates the conditional mean, whereas quantile regression estimates the conditional quantile of interest (e.g., 75th quantile or 50th quantile) [31,32,33,34,35]. For each of the studied equations (GFR_NMR_, CKD-EPI_2021Cr_ and CKD-EPI_2021CrCys_), mGFR distribution was calculated at given eGFR thresholds (45, 60 and 90 mL/min/1.73 m^2^). These thresholds were chosen as they represent GFR decision values for CKD staging [36]. The 95% prediction interval (PI) of mGFR for a given eGFR threshold (e.g., 30 mL/min/1.73 m^2^) was defined as the predicted mGFR by the 97.5th quantile model at eGFR equals 30 mL/min/1.73 m^2^ minus the predicted mGFR by the 2.5th quantile model at eGFR equals 30 mL/min/1.73 m^2^. This 95% PI is thus expected to include approximately 95% of the mGFR values from patients with a given eGFR.

## 3. Results

### 3.1. Patient Characteristics

A total of 120 sera were collected as part of routine clinical practice from patients whose mGFR was determined by urinary iothalamate clearance (Figure 1). GFR was estimated using three race-free eGFR equations: NMR-based GFR_NMR_ [11,12], guideline-recommended CKD-EPI_2021Cr_, and CKD-EPI_2021CrCys_ [3]. Of the 120 tested samples, 115 with a valid GFR_NMR_ result were included in the analysis (Figure 1).

Demographic and clinical characteristics of the study population (*n* = 115) are shown in Table 1. The study population included mainly White participants (97.4%), 56.5% were men, and mean (standard deviation [SD]) age was 54.9 (10.6) years. Most patients were solid-organ transplant recipients (88.7%, with 60% post kidney transplantation), had chronic kidney disease (CKD; 66.1%), and presented comorbidities, such as hypertension (69.6%) and dyslipidemia (65.2%). The majority (114/115 [99.1%]) of patients were on medications including immunosuppressive agents (87.8%), corticosteroids (47.8%), and beta blockers (35.7%) (Table 1).

Mean (SD) mGFR in the study population was 64.2 (20.8) mL/min/1.73 m^2^, and mean (SD) eGFR for GFR_NMR_, CKD-EPI_2021Cr_, and CKD-EPI_2021CrCys_ were 64.1 (18.7), 63.6 (20.2), and 63.8 (21.5) mL/min/1.73 m^2^, respectively (Table 1).

### 3.2. Performance of eGFR Equations in Routine Clinical Samples 

#### 3.2.1. Bias, Precision, and Accuracy

Median bias to mGFR of the three eGFR equations was overall low and slightly overestimated mGFR (positive median bias ranging from 0.4 to 2.0 mL/min/1.73 m^2^) (Table 2). Median bias of GFR_NMR_ was not statistically significantly different from that of the 2021 CKD-EPI equations (*p* > 0.05 in pairwise comparisons; Table 2). However, despite comparable median bias, the bias distribution of GFR_NMR_ differed from that of the CKD-EPI equations. GFR_NMR_ bias distribution was unimodal with one narrow peak centered around its median value of 2.0 mL/min/1.73 m^2^, while CKD-EPI_2021Cr_ and CKD-EPI_2021CrCys_ bias distribution were bimodal, with one peak of negative biases and another of positive biases (Figure 2). This heterogeneous bias distribution of eGFR determined by CKD-EPI_2021Cr_ and CKD-EPI_2021CrCys_ indicates that these equations often underestimated GFR, as opposed to GFR_NMR_. 

GFR_NMR_ showed a significantly higher precision than CKD-EPI_2021Cr_ (*p* = 0.01) and CKD-EPI_2021CrCys_ (*p* = 0.01), with an interquartile range (IQR) of the difference to mGFR of 10.5 mL/min/1.73 m^2^ (Table 2).

GFR_NMR_ accuracy ranged from 66.1% (P15) to 95.7% (P30) (Table 2), and was higher than that of both 2021 CKD-EPI equations at any error tolerance cutoffs (Table 2 and Figure 3). GFR_NMR_ statistically significantly outperformed CKD-EPI_2021Cr_ and CKD-EPI_2021CrCys_ equations regarding P15 and P30 accuracy (*p*-values between 0.006 and 0.02; Table 2). For P20 accuracy, GFR_NMR_ significantly outperformed CKD-EPI_2021CrCys_ (*p* = 0.001), but not CKD-EPI_2021Cr_ (80.0% vs. 73.0% for GFR_NMR_ vs. CKD-EPI_2021Cr_; *p* = 0.19) (Table 2).

#### 3.2.2. Prediction Intervals

For each of the studied equations (GFR_NMR_, CKD-EPI_2021Cr_ and CKD-EPI_2021CrCys_), we assessed the distribution of mGFR at eGFR thresholds used to define CKD stages, by determining the mGFR 95% prediction intervals (PI). The 95% PI is expected to include approximately 95% (2.5th to 97.5th percentiles) of the mGFR values from patients with a given eGFR. The width of the 95% PI has direct clinical implications on the precision of CKD staging [37]. Due to the small number (*n* = 3) of patients with GFR < 30 mL/min/1.73 m^2^ in our study population (Table 1, CKD stages G4 and G5), we considered only the eGFR thresholds of 45, 60, and 90 mL/min/1.73 m^2^. 

At the three investigated eGFR thresholds (45, 60 and 90 mL/min/1.73 m^2^), the width of the 95% PI, but also of the 50% PI, was consistently smaller for GFR_NMR_ compared to either CKD-EPI_2021Cr_ or CKD-EPI_2021CrCys_ (Figure 4). The predicted median mGFR (50th percentile) for GFR_NMR_ of 45, 60, and 90 mL/min/1.73 m^2^ was 44.7, 58.2, and 85.4 mL/min/1.73 m^2^, respectively. 

At a GFR_NMR_ of 45 mL/min/1.73 m^2^, 50% of mGFR ranged from 41.3 to 50.0 mL/min/1.73 m^2^, 80% from 37.5 to 58.6 mL/min/1.73 m^2^, and 95% from 34.3 to 61.7 mL/min/1.73 m^2^. 

At a GFR_NMR_ of 60 mL/min/1.73 m^2^, 50% of mGFR ranged from 54.1 to 65.0 mL/min/1.73 m^2^, 80% from 49.9 to 75.1 mL/min/1.73 m^2^, and 95% from 44.3 to 81.9 mL/min/1.73 m^2^. 

At a GFR_NMR_ of 90 mL/min/1.73 m^2^, 50% of mGFR ranged from 79.7 to 95.0 mL/min/1.73 m^2^, 80% from 74.6 to 108.1 mL/min/1.73 m^2^, and 95% from 64.3 to 122.4 mL/min/1.73 m^2^. Thus, at this eGFR threshold, the 95% PI did not cross the adjacent CKD stage cutoff of 60 mL/min/1.73 m^2^, in contrast to the 95% PI at the eGFR thresholds of 45 and 60 mL/min/1.73 m^2^, which crossed the next CKD stage cutoff (either p2.5 or p97.5 percentile) by no more than 1.7 mL/min/1.73 m^2^ (Figure 4).

### 3.3. Impact of Disease, Comorbidities and Medication on the Performance of eGFR Equations

To evaluate the impact of clinical parameters on the performance of the three eGFR equations, subgroup analyses were performed according to the documented etiology, comorbidities, and administered medications. This analysis was limited to groups of >25 samples and to conditions related to kidney function or dysfunction (Table 1). Bias distribution for each equation was compared between subgroups of patients with vs. without CKD, kidney transplantation, hypertension, or hyperlipidemia, and those treated vs. untreated with immunosuppressive agents, corticosteroids, or beta blockers (Table 3). 

Except for CKD-EPI_2021Cr_ and CKD-EPI_2021CrCys_ between immunosuppressive treatment groups (CKD-EPI_2021Cr_: median bias [95% CI] of 1.2 [−1.5; 5.5] vs. −11.8 [−21.1; −9.4]; *p* = 0.002; CKD-EPI_2021CrCys_: median bias [95% CI] of 1.9 [−1.0; 6.0] vs. −10.7 [−17.8; −5.6]; *p* = 0.007), and GFR_NMR_ between beta-blocker treatment groups (median bias [95% CI] of 0.0 [−3.0; 4.0] vs. 2.0 [0.0; 4.0]; *p* = 0.05), bias distribution between subgroups was not statistically significantly different (Table 3).

## 4. Discussion

The performance of three eGFR equations (GFR_NMR_, CKD-EPI_2021Cr_, and CKD-EPI_2021CrCys_) was compared using fresh samples of patients with clinically ordered urinary iothalamate clearance mGFR. The three equations showed an overall low median bias, however the bias distributions differed in subtle but potentially important ways. Bias distribution for GFR_NMR_ was centered around its median of 2.0 mL/min/1.73 m^2^, while that of the CKD-EPI equations was broad and bimodal, indicating heterogeneity in performance. This heterogenicity may be reflective of the different confounders for creatinine and cystatin C, which can be in opposite directions. On the other hand, GFR_NMR_ outperformed the 2021 CKD-EPI equations in terms of precision (interquartile range to mGFR of 10.5 mL/min/1.73 m^2^) and accuracy (P15, P20, and P30 of 66.1%, 80.0%, and 95.7%, respectively).

Similar to the 2021 CKD-EPI equations, GFR_NMR_ is a race-free equation, and like CKD-EPI_2021CrCys_, GFR_NMR_ integrates both creatinine and cystatin C as biomarkers in addition to age and sex. The improved precision and accuracy of GFR_NMR_ compared to CKD-EPI_2021CrCys_ supports the benefit of adding myo-inositol and valine to the eGFR equation. These results further support published reports and recommendations on the need for adequate biomarker constellations to improve GFR estimation [9,10,11,17,38,39,40].

The GFR_NMR_ performance evaluation reported here confirms in an independent cohort the findings of our first validation study [11]. Both studies, employing an iothalamate-based mGFR reference standard and including patients with comparable demographics (Mayo Clinic, Rochester, NY, USA), confirmed the results obtained for GFR_NMR_ as to precision (10.5 vs. 13.0 mL/min/1.73 m^2^ in the present vs. previous study) and accuracy (66.1% vs. 61.2% [P15], 80.0% vs. 71.5% [P20], 95.7% vs. 87.2% [P30], in the present vs. previous study). This finding further supports the suitability of GFR_NMR_ for use in clinical routine settings, particularly for e.g., the accurate classification of CKD stages, assessment of renal impairment and renal drug dose adjustment.

For the first time, we were able to systematically investigate the impact of a variety of etiologies, comorbidities, and medications on GFR_NMR_ results. Although clinical data were collected on a wide range of conditions, the statistical analysis had to be restricted to groups of > 25 samples and to univariate statistics to ensure result integrity. While in most comparisons, no statistically significant differences in bias distributions were found between groups, we observed a significant difference in bias for GFR_NMR_ between beta-blocker treatment groups (median bias [95% CI] of 0.0 [−3.0; 4.0] vs. 2.0 [0.0; 4.0]; *p* = 0.05). In the kidney, adrenergic receptors mediate vasoconstriction, renin secretion, and vasodilation, respectively [41,42,43]. Blockade of beta-receptors may therefore affect renal blood flow and possibly GFR through intrarenal effects [41,42,43]. Interestingly, all investigated eGFR equations tended to underestimate mGFR in the beta-blocker treatment group compared to the control group, an observation consistent with a decreased renal perfusion. However, the univariate nature of our evaluation due to the small sample size prevented us from correcting the observed effect sizes for important covariates, such as age, sex, race, other medications, or CKD etiology. Significantly larger patient populations for such an analysis is warranted. 

Limitations of our study include its small population size (*n* = 115), its single center nature, the high proportion (88.7%) of solid-organ transplant recipients, the small proportion (<3%) of non-White participants, and the small number (*n* = 3) of patients with low (<30 mL/min/1.73 m^2^) mGFR. The latter is critical because low mGFR thresholds represent essential clinical decision points, such as to eligibility to kidney transplantation listing (at GFR < 20 mL/min/1.73 m^2^ in the US). The weak representation of non-White participants and of patients with low mGFR, and the strong representation of certain medical conditions (such as solid-organ transplantation) constitute the drawback of a real-world single-center study. Future real-world studies should be conducted at multiple centers to cover more ethnicities, more patients with severe CKD, and more diverse medical conditions.

Strengths of our study include its real-world setting, the subgroup analysis according to comorbidities and medication, and the use of a single standardized method for mGFR determination (urinary iothalamate clearance). This is important because the use of mixed mGFR determination methods might introduce a bias in eGFR performance interpretation [3,11,14,44,45]. In fact, recent studies evaluating GFR_NMR_ exclusively relied on urinary iothalamate clearance (this study and [13]).

Overall, our results add real-world eGFR validation data to the growing body of evidence confirming the suitability of GFR_NMR_ to meet the NKF-ASN Task Force recommendations, and hold promise for an alternative, well-performing, and equitable eGFR equation.

## 5. Conclusions

This real-world study demonstrated a superior performance of GFR_NMR_ compared to the CKD-EPI_2021Cr_ and CKD-EPI_2021CrCys_ equations on routine samples. This study thus validates the use of GFR_NMR_ for accurate estimation of GFR in routine clinical practice, potentially improving patient management.

## Figures and Tables

**Figure 1 bioengineering-10-00717-f001:**
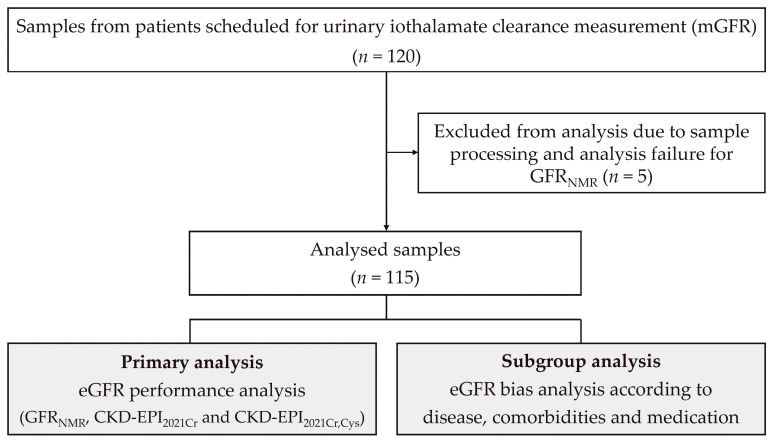
Study flow diagram. Abbreviations: CKD-EPI, Chronic Kidney Disease—Epidemiology Collaboration; CKD-EPI_2021Cr_, 2021 creatinine eGFR equation without race [3]; CKD-EPI_2021CrCys_, 2021 creatinine-cystatin C eGFR equation without race [3]; eGFR, estimated GFR; GFR, glomerular filtration rate; GFR_NMR_, NMR-based eGFR equation [11,12]; mGFR, measured GFR.

**Figure 2 bioengineering-10-00717-f002:**
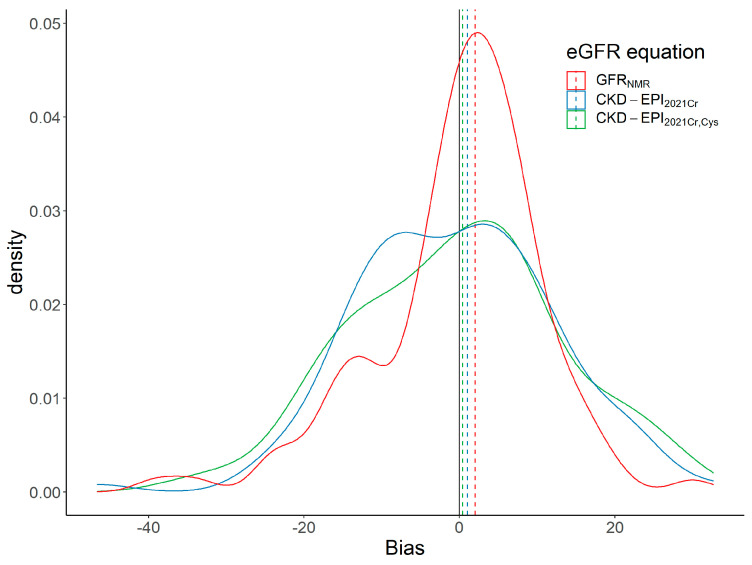
Bias distribution for GFR_NMR_, CKD−EPI_2021Cr_, and CKD−EPI2021_CrCys_. The dashed lines indicate the median bias. GFR_NMR_ bias distribution appeared unimodal, with one main peak centered around its median value of 2.0 mL/min/1.73 m^2^. By contrast, CKD-EPI_2021Cr_ and CKD−EPI2021_CrCys_ bias distribution appeared bimodal, with a pool of patients with negative bias and another with positive bias. Median bias of all three equations was positive and close to zero (Table 2). Abbreviations: CKD−EPI_2021Cr_, 2021 creatinine eGFR equation without race [3]; CKD−EPI_2021CrCys_, 2021 creatinine−cystatin C eGFR equation without race [3]; eGFR, estimated GFR; GFR, glomerular filtration rate; GFR_NMR_, NMR−based eGFR equation [11,12].

**Figure 3 bioengineering-10-00717-f003:**
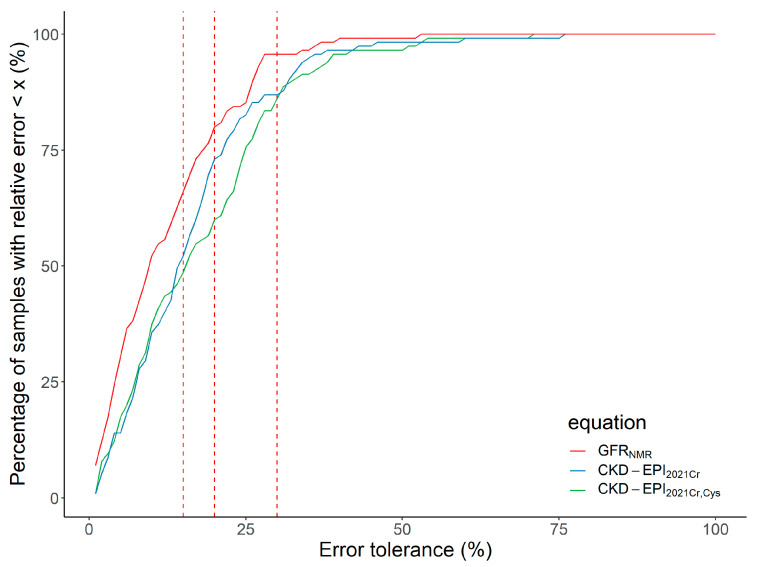
Regression Error Characteristic Curve for all error tolerance Px values for GFR_NMR_, CKD-EPI_2021Cr_, and CKD-EPI2021_CrCys_. Px denotes the percentage of eGFR values within x% of mGFR. The red dashed lines indicate error tolerance cutoffs of (from left to right) P15, P20, and P30, respectively. The y-axis shows the corresponding percentage of samples within the given error tolerance Px value on the x-axis. Abbreviations: CKD-EPI_2021Cr_, 2021 creatinine eGFR equation without race [3]; CKD-EPI_2021CrCys_, 2021 creatinine-cystatin C eGFR equation without race [3]; eGFR, estimated GFR; GFR, glomerular filtration rate; GFR_NMR_, NMR-based eGFR equation [11,12].

**Figure 4 bioengineering-10-00717-f004:**
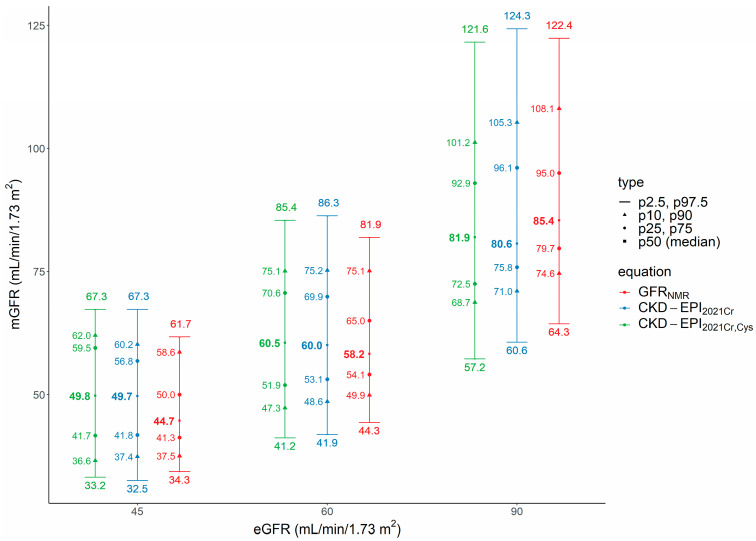
Distribution of mGFR at selected eGFR thresholds (45, 60 and 90 mL/min/1.73 m^2^). The whiskers represent the 95% prediction intervals (PI) between mGFR and eGFR (2.5th to 97.5th percentiles; p2.5, p97.5). The symbols (triangles, circles, and squares) indicate the percentiles of mGFR at a given eGFR (p10 to p90, p25 to p75, and p50 or median, respectively). At a given eGFR, 50% of mGFR values range from the 25th to 75th percentiles, 80% from the 10th to 90th percentiles, and 95% from the 2.5th to 97.5th percentiles (defined as 95% PI). Abbreviations: CKD-EPI_2021Cr_, 2021 creatinine eGFR equation without race [3]; CKD-EPI_2021CrCys_, 2021 creatinine-cystatin C eGFR equation without race [3]; eGFR, estimated GFR; GFR, glomerular filtration rate; GFR_NMR_, NMR-based eGFR equation [11,12]; mGFR, measured GFR; px, percentile.

**Table 1 bioengineering-10-00717-t001:** Patient characteristics.

Study Population, N (%)	115 (100.0%)
Sex, N (%)	
Female	50 (43.5%)
Male	65 (56.5%)
Age group in years, N (%)	
<50	37 (32.2%)
50–64	55 (47.8%)
≥65	23 (20.0%)
Age in years, Mean ± SD (range)	54.9 ± 10.6 (32.0–78.0)
Height in cm, Mean ± SD (range) ^1^	171.7 ± 9.9 (148.0–195.0)
Weight in kg, Mean ± SD (range) ^1^	87.9 ± 20.8 (51.0–161.0)
Ethnicity, N (%)	
White	112 (97.4%)
Asian	1 (0.9%)
Black or African American	1 (0.9%)
Not disclosed	1 (0.9%)
CKD Stage, N (%) ^2^	
G1	13 (11.3%)
G2	48 (41.7%)
G3a	36 (31.3%)
G3b	15 (13.0%)
G4	3 (2.6%)
G5	0 (0.0%)
Underlying disease or condition, N (%)	
Solid-organ transplantation (n missing = 13) ^1^	102 (88.7%)
Kidney transplantation ^3^	69 (60.0%)
Chronic kidney disease (CKD) (n missing = 3) ^1^	76 (66.1%)
Liver disease	33 (28.7%)
Nephrectomy (n missing = 1) ^1^	13 (11.3%)
Concomitant disease, N (%)	
Hypertension	80 (69.6%)
Dyslipidemia	75 (65.2%)
Hyperlipidemia	67 (60.0%)
Diabetes mellitus (n missing = 1) ^1^	26 (22.6%)
Cardiovascular disease (n missing = 2) ^1^	22 (19.1%)
Medication, N (%)	
Immunosuppressive agents (n missing = 1) ^1^	101 (87.8%)
Corticosteroids (n missing = 2) ^1^	55 (47.8%)
Beta-blocker (n missing = 2) ^1^	41 (35.7%)
ACE inhibitor (n missing = 2) ^1^	21 (18.3%)
Antidiabetics (n missing = 2) ^1^	22 (19.1%)
Measured GFR (mGFR), Mean (SD) ^4^	64.2 (20.8)
Estimated GFR (eGFR), Mean (SD) ^4^	
CKD-EPI2021Cr	63.6 (20.2)
CKD-EPI2021CrCys	63.8 (21.5)
GFRNMR	64.1 (18.7)

^1^ Percentages refer to documented characteristics in the study population, not taking into account patients with missing characteristics (the number of patients with missing characteristics is indicated in brackets); ^2^ CKD staging based on mGFR and according to the Kidney Disease: Improving Global Outcomes (KDIGO) guideline [36]; ^3^ Includes patients with single kidney transplantation, combined kidney-liver or kidney-pancreas transplantation; ^4^ Expressed as mL/min/1.73 m^2^ of body-surface area. Abbreviations: ACE, angiotensin-converting enzyme; CKD, chronic kidney disease; CKD-EPI, Chronic Kidney Disease—Epidemiology Collaboration; CKD-EPI_2021Cr_, 2021 creatinine eGFR equation without race [3]; CKD-EPI_2021CrCys_, 2021 creatinine-cystatin C eGFR equation without race [3]; eGFR, estimated GFR; GFR, glomerular filtration rate; GFR_NMR_, NMR-based eGFR equation [11,12]; mGFR, measured GFR; N, number of samples; SD, standard deviation.

**Table 2 bioengineering-10-00717-t002:** Performance of eGFR equations.

Variable	Performance Value
Median signed bias to mGFR (95% CI) ^1^	
CKD-EPI_2021Cr_	1.0 (0.3; 6.8)
CKD-EPI_2021CrCys_	**0.4** (−2.1; 3.3)
GFR_NMR_	2.0 (1.0; 4.0)
Precision—IQR of the difference to mGFR (95% CI) ^1^	
CKD-EPI_2021Cr_	16.8 (12.9; 19.9) *
CKD-EPI_2021CrCys_	17.9 (14.0; 21.9) *
GFR_NMR_	**10.5** (5.5; 13.0)
Accuracy—P15 (95% CI) [%] ^2^	
CKD-EPI_2021Cr_	52.2 (43.5; 60.9) *
CKD-EPI_2021CrCys_	48.7 (39.2; 57.4) **
GFR_NMR_	**66.1** (57.4; 74.8)
Accuracy—P20 (95% CI) [%] ^2^	
CKD-EPI_2021Cr_	73.0 (64.3; 80.9)
CKD-EPI_2021CrCys_	60.0 (51.3; 68.7) ***
GFR_NMR_	**80.0** (73.0; 87.0)
Accuracy—P30 (95% CI) [%] ^2^	
CKD-EPI_2021Cr_	87.0 (80.9; 93.0) *
CKD-EPI_2021CrCys_	86.1 (80.0; 92.2) **
GFR_NMR_	**95.7** (92.2; 100.0)

^1^ Expressed as mL/min/1.73 m^2^; ^2^ P15, P20, and P30 denote the percentage of eGFR values lying within the tolerance range of 15%, 20%, and 30% of measured GFR (mGFR), respectively. Bold numbers highlight the best performance results in each analysis. Symbols *, ** and *** indicate the level of significance for *p*-values < 0.05, < 0.01, and < 0.001, respectively, in the pairwise tests against GFR_NMR_ for each KPI. Abbreviations: CI, confidence interval; CKD-EPI, Chronic Kidney Disease—Epidemiology Collaboration; CKD-EPI_2021Cr_, 2021 creatinine eGFR equation without race [3]; CKD-EPI_2021CrCys_, 2021 creatinine-cystatin C eGFR equation without race [3]; eGFR, estimated GFR; GFR, glomerular filtration rate; GFR_NMR_, NMR-based eGFR equation [11,12]; IQR, interquartile range. mGFR, measured GFR.

**Table 3 bioengineering-10-00717-t003:** Comparison of median signed bias (95% CI) for GFR_NMR_, CKD-EPI_2021Cr_, and CKD-EPI_2021CrCys_ according to underlying disease, comorbidities, and medication.

Variable		eGFR Equation		
		GFR_NMR_	CKD—EPI_2021Cr_	CKD—EPI_2021CrCys_
CKD	Yes (*n* = 76)	0.5 (−2.0; 2.0)	1.3 (−1.2; 5.8)	2.1 (−0.8; 6.4)
	No (*n* = 39)	2.0 (−1.0; 7.0)	−6.9 (−15.6; −2.6)	−2.4 (−7.9; 5.2)
	*p*-value ^1^	0.96	0.12	0.29
Kidney transplantation	Yes (*n* = 69)	1.0 (−1.0; 3.0)	1.5 (−1.1; 6.1)	2.3 (−1.3; 6.8)
	No (*n* = 46)	2.0 (−1.0; 5.0)	−6.4 (−14.7; −2.4)	−2.2 (−6.6; 5.5)
	*p*-value ^1^	0.94	0.12	0.18
Hypertension	Yes (*n* = 80)	0.5 (−2.0; 2.0)	1.1 (−0.9; 5.4)	0.6 (−2.7; 4.4)
	No (*n* = 35)	2.0 (−2.0; 4.0)	−4.7 (−13.5; −2.3)	−0.9 (−8.4; 2.8)
	*p*-value ^1^	0.33	0.78	0.79
Hyperlipidemia	Yes (*n* = 67)	0.0 (−3.0; 1.0)	−1.0 (−4.7; 4.0)	−0.8 (−5.2; 2.0)
	No (*n* = 48)	2.0 (−1.0; 4.5)	1.7 (−0.4; 9.5)	0.9 (−5.1; 5.7)
	*p*-value ^1^	0.41	0.64	0.50
Immunosuppressive agents	Yes (*n* = 101)	2.0 (1.0; 4.0)	1.2 (−1.5; 5.5)	1.9 (−1.0; 6.0)
	No (*n* = 14)	0.5 (−5.0; 13.0)	−11.8 (−21.1; −9.4)	−10.7 (−17.8; −5.6)
	*p*-value ^1^	0.21	**0.002**	**0.007**
Corticosteroids	Yes (*n* = 55)	1.0 (−1.0; 3.0)	1.6 (−3.1; 6.4)	2.3 (−0.8; 6.8)
	No (*n* = 60)	2.0 (0.0; 5.0)	−3.7 (−9.2; −0.3)	−2.0 (−6.9; 3.8)
	*p*-value ^1^	0.88	0.11	0.13
Beta blockers	Yes (*n* = 41)	0.0 (−3.0; 4.0)	−0.2 (−3.6; 6.6)	−2.7 (−10.9; 2.1)
	No (*n* = 74)	2.0 (0.0; 4.0)	1.1 (−1.6; 7.2)	1.2 (−1.7; 4.5)
	*p*-value ^1^	**0.05**	0.43	0.22

^1^ The Wilcoxon rank sum test was used to compare bias distributions in patients with (‘Yes’) vs. without (‘No’) the indicated disease, comorbidity, or medication. Bold *p*-values indicate statistical significance.

## Data Availability

The data presented in this study are available within the article.

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
