# Peer review of "Performance of Nuclear Magnetic Resonance-Based Estimated Glomerular Filtration Rate in a Real-World Setting"

_bioengineering, 2023, doi:10.3390/bioengineering10060717_

Round 1

Reviewer 1 Report

The manuscript by Santamaria et al. evaluates the performance of a new formula for the estimated Glomerular Filtration Rate in a small single-center cohort of patients scheduled for urinary iothalamate clearance measurement at the Mayo Clinic.

This is a follow-up study using an incident cohort of 115 patients, aiming at the validation of a new technique for the estimation of the GFR (AXINON® GFR (NMR) - https://www.numares.com/products/nephrology/axinonreggfr-nmr#axinonreggfr-nmr, available in Europe but without the FDA authorization for the USA) based on nuclear magnetic resonance already presented in previous papers.

The study sample is less than adequate to test the research hypothesis; on the other hand, the statistical analysis is sound in almost every part.

The major limitation of the manuscript concern the presentation of the study and its reproducibility.

First of all, the presentation of the results as a principal analysis (whole population) and a subgroup analysis (patients with mGFR ≤ 90) could be misleading: the difference in terms of sample numerosity (115 vs 103) is too small to justify the presentation of separate results (in fact the results between the two “groups” are much the same). Please present results only for the whole population.

Moreover, the definition of the GFRNMR as a formula could be misleading. First of all, the formula should be presented at least as supplementary material as in Table 2 of [Stämmler et al. Diagnostics 2021]. Then, the authors should clarify whether this formula is independent of the methods to measure the metabolite. In other words, is it possible to calculate the new formula using a different method to measure serum creatinine, valine, and myo-inositol, or is it applicable only using AXINON®GFR (NMR)? I think it is safe to say that in the near future NMR technology won’t be available in the vast majority of clinical laboratories, so the authors should discuss the possible cost-efficacy implication of introducing NMR in clinical practice for the relatively small improvement ensured by the new formula, or, on the other hand, whether using classical (cheaper) laboratory methods (e.g. chromatography) to measure valine and myo-inositol, the new formula could be effectively tested in a “real-world setting”.

About the statistical analyses presented, the use of NRI, especially in the case of non-nested models, as reclassification statistics has been criticized and it has been proven inferior to more classical tests. Please refer to [https://www.ncbi.nlm.nih.gov/pmc/articles/PMC3918180/, https://pubs.rsna.org/doi/10.1148/radiol.222343, https://www.ncbi.nlm.nih.gov/pmc/articles/PMC6568208/, https://www.fharrell.com/post/addvalue/#case-study-quantifying-diagnostic-information] and consider using other tests.

Minor concerns: please include the use of a majority of transplanted patients in the limitations of the study.

Author Response

First of all, the presentation of the results as a principal analysis (whole population) and a subgroup analysis (patients with mGFR ≤ 90) could be misleading: the difference in terms of sample numerosity (115 vs 103) is too small to justify the presentation of separate results (in fact the results between the two “groups” are much the same). Please present results only for the whole population.

Response: The manuscript was revised, as requested by Reviewer 1. Only the results for the whole population are now presented.

Moreover, the definition of the GFRNMR as a formula could be misleading. First of all, the formula should be presented at least as supplementary material as in Table 2 of [Stämmler et al. Diagnostics 2021].

Response: The formula for GFRNMR is clearly referenced in the manuscript, and hence simple to locate in the supplement of Stämmler et al. 2021. We believe that repeating the formula in the current manuscript could add redundant publishing of scientific results and be misleading or confusing for the reader, who might think that this formula differs from the one originally described in Stämmler et al. 2021. Therefore, we did not repeat the presentation of the formula in the revised manuscript. We sincerely hope the editorial team will agree with this decision.

Then, the authors should clarify whether this formula is independent of the methods to measure the metabolite. In other words, is it possible to calculate the new formula using a different method to measure serum creatinine, valine, and myo-inositol, or is it applicable only using AXINON®GFR (NMR)?

Response: The reviewer touches on a very interesting issue. In Stämmler et al. 2021 we explicitly chose to present the GFRNMR equation using calibrated metabolite concentrations in µmol/L rather than NMR signal integral values to make the approach independent of the NMR platform (see also Fuhrmann at al. 2022 for analytical validation). Thus, in theory, any analytical platform can be used to quantify the metabolites and use the published equation to calculate the GFRNMR test result. However, to the best of our knowledge, this has not yet been validated and reported. For practical purposes, this would require extensive analytical validation to ensure that the non-NMR quantifications do not have a systematic bias compared to the NMR-derived values; perhaps one reason why this has not yet been done. Another interesting question would be the imprecision behavior of such a "patchwork" approach. Since NMR simultaneously quantifies all metabolites in a single physical measurement, any instrumental imprecision applies to the metabolites in the same way, i.e. the relationship to each other remains intact. This would not be the case with independent chemical assays. Here each marker would show its own imprecision behavior. What this means for the imprecision of a "patchwork" test result compared to the original GFRNMR test result is unclear. We suspect that the imprecisions may add up. In conclusion, this certainly warrants further investigation and may help to overcome the barriers of test availability in routine clinical scenarios. However, this is clearly beyond the scope of our current work presented here. As such a discussion would also require extensive background, we feel that adding such a paragraph may overwhelm the interested reader. We hope that our detailed point-by-point response will be deemed sufficient by both the esteemed reviewer and the editorial team.

I think it is safe to say that in the near future NMR technology won’t be available in the vast majority of clinical laboratories, so the authors should discuss the possible cost-efficacy implication of introducing NMR in clinical practice for the relatively small improvement ensured by the new formula, or, on the other hand, whether using classical (cheaper) laboratory methods (e.g. chromatography) to measure valine and myo-inositol, the new formula could be effectively tested in a “real-world setting”.

Response: The reviewer is pointing to a crucial obstacle of all medical innovations: Accessibility. Low-threshold accessibility of innovations is indispensable for widespread use in a healthcare system. For assessment of renal functional reserve in patients with chronic (non-acute) kidney disease, a time to result in days rather than in hours is compatible with clinical routine. This opens the possibility to ship sample material to nationally operating central reference laboratories, providing overnight services. In the cardiology space, the NMR platform itself has already been successfully applied in such a routine setting for more than a decade to provide millions of low-density lipoprotein particle concentrations for advanced cardiovascular risk assessment. We believe this might represent a suitable blueprint for GFRNMR as well. We highlighted this in one of our recent papers (see. Stämmler et al. BMC Nephrol. 2023). The aim of this manuscript was to present the scientific and clinical benefits of GFRNMR in a routine clinical setting. Given that this current test is not yet commercially available, and reimbursement rates have not been specified, a sound cost-efficiency analysis at present remains difficult and premature. We agree with the reviewer that cost-efficiency should be investigated in a future study.

About the statistical analyses presented, the use of NRI, especially in the case of non-nested models, as reclassification statistics has been criticized and it has been proven inferior to more classical tests. Please refer to [https://www.ncbi.nlm.nih.gov/pmc/articles/PMC3918180/, https://pubs.rsna.org/doi/10.1148/radiol.222343, https://www.ncbi.nlm.nih.gov/pmc/articles/PMC6568208/, https://www.fharrell.com/post/addvalue/#case-study-quantifying-diagnostic-information] and consider using other tests.

Response: We agree with Reviewer 1 that the interpretation of the NRI can be misleading as it tends to overemphasize differences between models. After considering alternative testing methods, and failing to identify an appropriate test compatible with our data, i.e. with five categories (CDK stages) and no prediction probabilities for each of these categories, we decided to removed the NRI data from the revised manuscript.

Minor concerns: please include the use of a majority of transplanted patients in the limitations of the study.

Response: Following the reviewer’s request, we now included the high proportion of solid-organ transplant recipients in the limitations of the study. The text now reads:

Limitations of our study include its small population size (n = 115), its single center nature, the high proportion (88.7%) of solid-organ transplant recipients, the small proportion (< 3%) of non-White participants, and the small number (n = 3) of patients with low (< 30 mL/min/1.73 m2) mGFR. The latter is critical because low mGFR thresholds represent essential clinical decision points, such as to eligibility to kidney transplantation listing (at GFR < 20 mL/min/1.73 m2 in the US). The weak representation of non-White participants and of patients with low mGFR, and the strong representation of certain medical conditions (such as solid-organ transplantation) constitute the drawback of a real-world single-center study. Future real-world studies should be conducted at multiple centers as to cover more ethnicities, more patients with severe CKD, and more diverse medical conditions.” 

Reviewer 2 Report

In this study, the authors systematically investigate the impact of a variety of etiologies, comorbidities, and medications on GFRNMR results. The GFRNMR equation combining the serum creatinine, cystatin C, valine, and myo-inositol gave us a more accurate evaluate method. The equation could be validated in more populations before using in the clinical setting.

Author Response

Response: We thank Reviewer 2 for the comment. It is indeed our intention to widen the study population and validate the test on a variety of subjects to ensure the test has a sound generalizability in a clinical setting.

Reviewer 3 Report

The study from Santamaria et al. is very well performed and clearly presented. The validation of GFRNMR method in a setting of routine clinical practice, despite the limitation to have a small proportion of non-white participants and patients with low (< 30 mL/min/1.73 m2) mGFR,  adds relevant value to the previous studies where the method was first described. In my opinion, the paper deserves publication in Bioengineering in its present form.

Author Response

Response: We thank Reviewer 3 for the positive evaluation of our study.

Round 2

Reviewer 1 Report

The manuscript has been improved as requested